# Fundamental Results of Cyclic Codes over Octonion Integers and Their Decoding Algorithm

**Muhammad Sajjad** [1,*] , **Tariq Shah** [1], **Robinson-Julian Serna** [2] , **Zagalo Enrique Suárez Aguilar** [2] **and Omaida Sepúlveda Delgado** [2]

1 Department of Mathematics, Quaid-I-Azam University, Islamabad 45320, Pakistan
2 Escuela de Matemáticas y Estadística, Universidad Pedagógica y Tecnológica de Colombia, Avenida Central del Norte 39-115, Tunja 150003, Colombia
* Correspondence: m.sajjad@math.qau.edu.pk

**Abstract:** Coding theory is the study of the properties of codes and their respective fitness for specific applications. Codes are used for data compression, cryptography, error detection, error correction, data transmission, and data storage. Codes are studied by various scientific disciplines, such as information theory, electrical engineering, mathematics, linguistics, and computer science, to design efficient and reliable data transmission methods. Many authors in the previous literature have discussed codes over finite fields, Gaussian integers, quaternion integers, etc. In this article, the author defines octonion integers, fundamental theorems related to octonion integers, encoding, and decoding of cyclic codes over the residue class of octonion integers with respect to the octonion Mannheim weight one. The comparison of primes, lengths, cardinality, dimension, and code rate with respect to Quaternion Integers and Octonion Integers will be discussed.

**Keywords:** octonion integers; octonion Mannheim distance; cyclic codes; encoding; syndromes decoding

## 1. Introduction

Coding theory is a branch of mathematics that has many applications in information theory. Many types of codes and their parameters have been extensively studied. As one of the essential parameters of the code, the distance-related (Hamming, Lee, Mannheim, and so on) are also examined for many types of codes, and the expression for the minimum or maximum values of distances have been found [1,2].

A portion of these codes, which have gone through huge improvement in recent years, are Integer Codes. Integer Codes will be codes characterized over limited rings of whole numbers modulo $m$, $m \in \mathbb{Z}$, and enjoy a few upper hands over the customary block codes. One of these benefits is that integer codes are fit for rectifying a predetermined number of blunder designs which happen most often, while the ordinary codes plan to address all conceivable error designs, without completely succeeding. In real communication systems, integer codes have a low encoding and decoding complexity for a suitable application [3]. There are a few different codes like the Integer Codes, such as codes over Gaussian integers, Eisenstein–Jacobi integers [4–6], a class of mistake-revising codes given a summed-up Lee distance [7], codes over Hurwitz numbers [8], and so forth, which have been seriously contemplated as of late.

Quadrature amplitude modulation is used in various digital data radio communications and data communication applications. The most well-known errors which show up in numerous advanced information radio correspondences and information correspondence applications are those which change a point into its closest neighbor. The Hamming and the Lee distance cannot correct these errors in a QAM signal. In [5], Huber developed codes over Gaussian integers with another distance to advance the present circumstance, called the Mannheim distance. He demonstrated that these codes could address the Mannheim error of weight 1 and utilized this new distance to track down the properties of these codes

(see [9] for additional subtleties). Furthermore, in [7], the authors presented another distance that summed up the Lee distance and built codes fit for adjusting errors of summed up Lee weight one or two.

Notwithstanding, Neto et al. [10] got cyclic codes over Euclidean quadratic field $Q\left[\sqrt{d}\right]$, where $d = -1, -2, -3, -7, -11$, by utilizing the Mannheim metric. They additionally offered decoding procedures to address errors of any Mannheim weight upsetting one and two coordinates of a cyclic code vector. Cyclic codes over finite rings regarding the Mannheim metric were acquired by involving Gaussian integers in [10]. Afterward, in [11], Özen and Güzeltepe utilized quaternion Mannheim metric perfect codes over finite quaternion rings were gotten and decoded these codes. Özen and Güzeltepe [12] treated the error of amendment of cyclic codes over quaternion numbers for quaternion Mannheim weight one. Shah and Rasool [13] got the decoding algorithms for the correction of errors of quaternion Mannheim weight two. Özen and Güzeltepe [14] got the cyclic codes over finite quaternion integer rings. Shah and Khan have constructed codes over semi-group-ring in [15]. Sajjad and Shah got quaternion integers based on higher-length cyclic codes and their decoding algorithm in [16].

In the present paper, the author defines octonion integers, fundamental theorems related to octonion integers, encoding, and decoding of cyclic codes over the residue class of octonion integers with respect to the octonion Mannheim weight one. The comparison of primes, lengths, cardinality, dimension, and code rate with respect to Quaternion Integers and Octonion Integers will be discussed.

This article is structured in the following Sections: In Section 2, octonion integers and some important algebraic notions are discussed. In Section 3, cyclic codes over octonion integer rings regarding the Octonion Mannheim metric are deliberated. Theorem 3 shows how to construct cyclic codes by using Proposition 1 and Theorem 2. In Proposition 3, the algebraic background is shown, which is important for constructing cyclic codes over the finite rings. Theorem 4, shows how to construct cyclic codes over the finite rings, and at the end decoding of the cyclic codes over octonion integers of octonion Mannheim weight one. A comparison of the proposed work with some existing works is described in Section 4. The conclusion and future directions are given in Section 5.

## 2. Octonion Integers

**Definition 1.** *An octonion algebra $O(\mathbb{R})$ over the set of real numbers $(\mathbb{R})$ is non-associative until algebra by the following conditions:*

- $O(\mathbb{R}) = \left\{ a_0 + \sum_{j=1}^{7} a_j i_j : a_j \in \mathbb{R} \right\}$ *is free $\mathbb{R}$-module over $1, i_1, i_2, i_3, i_4, i_5, i_6$ and $i_7$.*
- *1 is a Multiplicative unit.*
- $i_1{}^2 = i_2{}^2 = i_3{}^2 = i_4{}^2 = i_5{}^2 = i_6{}^2 = i_7{}^2 = -1$, $i_j i_k = -i_k i_j = i_q$, $j \neq k$, $j, k \in \{1, 2, 3, 4, 5, 6, 7\}$, *where, $q = j \otimes k$ is " $\times -or$" for $j$, $k$.*

The set $O(\mathbb{Z}) = \left\{ a_0 + \sum_{j=1}^{7} a_j i_j : a_j \in \mathbb{Z} \right\}$ is a subset of octonion algebra $O(\mathbb{R})$. If $o = a_0 + \sum_{j=1}^{7} a_j i_j$ is an octonion integer, then its octonion conjugate is $\bar{o} = a_0 - \sum_{j=1}^{7} a_j i_j$. The norm of $o$ is $N(o) = p = o\bar{o} = a_0^2 + a_1^2 + a_2^2 + a_3^2 + a_4^2 + a_5^2 + a_6^2 + a_7^2$. An octonion integer contains two parts: one is the real part and the other is the imaginary part. Let $o = a_0 + \sum_{j=1}^{7} a_j i_j$ be an octonion integer, then its real part is $a_0$, and the imaginary part is $\sum_{j=1}^{7} a_j i_j$. The commutative and associative property of multiplication does not hold for octonion integers. The commutative and associative property of multiplication for octonion integers holds only if the imaginary part of two octonion integers is parallel to each other. Define $O(K)$:

$$O(K) = \left\{ c + d \left( \sum_{j=1}^{7} i_j \right) : c, d \in \mathbb{Z} \right\},$$

Contains octonion integers. Thus, the commutative and associative property of multiplication holds for $O(K)$.

Theorems 1 and 2 show the relation between octonion integers and prime integers.

**Theorem 1.** *In [16], (Theorem 2.5.9), For every odd, rational prime $p \in N$, there exists a prime $\delta \in O(\mathbb{Z})$, such that $N(\delta) = p = \delta\bar{\delta}$. In particular, p is not prime in $O(\mathbb{Z})$.*

**Theorem 2.** *In [17,18], (Theorem 2.5.10), Let $\delta$ be the element of $O(\mathbb{Z})$ is a prime in $O(\mathbb{Z})$ if and only if $N(\delta)$ is prime in $\mathbb{Z}$.*

### 3. Cyclic Codes Based on Octonion Integers

The following Theorem is the modification of quaternion integers [15] to octonion integers.

**Theorem 3.** *If c and d are two integers and are relatively prime to each other, then $O(K)/< c + d\left(\sum_{j=1}^{7} i_j\right) >$ is isomorphic to $\mathbb{Z}_{c^2+7d^2}$.*

**Proof.** We can suppose without loss of generality $c$ and $d$ are positive integers. Then, the greatest common divisor of $b$ and $\mathbb{Z}_{c^2+7d^2}$ is 1, thus, $d^{-1}$ exists in $\mathbb{Z}_{c^2+7d^2}$. Since $c^2 + 7d^2 \equiv 0 (mod\ c^2 + 7d^2)$, $c^2 \equiv -7d^2 (mod\ c^2 + 7d^2)$, then, $(cd^{-1})^2 \equiv -7 (mod\ c^2 + 7d^2)$. Define $\theta : O(K) \to \mathbb{Z}_{c^2+7d^2}$ by $\theta\left(x + y\left(\sum_{j=1}^{7} i_j\right)\right) = x - (cd^{-1})y (mod\ c^2 + 7d^2)$, $\theta$ is surjective and preserves addition. $\square$

Let $\beta_1 = x_1 + y_1 \sum_{j=1}^{7}(i_j)$ and $\beta_2 = x_2 + y_2 \sum_{j=1}^{7}(i_j)$.
Since

$$
\begin{aligned}
\theta(\beta_1)\theta(\beta_2) &= \left(x_1 + (cd^{-1})y_1\right)\left(x_2 + (cd^{-1})y_2\right) \\
&\equiv \left(x_1 x_2 + \left(cd^{-1}\right)^2 y_1 y_2\right) - \left(cd^{-1}\right)(y_1 x_2 + y_2 x_1) \\
&\equiv (x_1 x_2 - 7y_1 y_2) - (cd^{-1})(y_1 x_2 + y_2 x_1) \\
&= \theta\left((x_1 x_2 - 7y_1 y_2) + (y_1 x_2 + y_2 x_1) \sum_{j=1}^{7}(i_j)\right) \\
&= \theta\left(x_1 + y_1 \sum_{j=1}^{7}(i_j)\right)\left(x_2 + y_2 \sum_{j=1}^{7}(i_j)\right) \\
&= \theta(\beta_1 \beta_2)
\end{aligned}
$$

$\theta$ Observes multiplication. Moreover, because $c + d\sum_{j=1}^{7}(i_j) = c - (cd^{-1})d \equiv 0$, $c + d\sum_{j=1}^{7}(i_j) \subseteq Ker(\theta)$, where $< . >$ represents an ideal generated by $c + d\sum_{j=1}^{7}(i_j)$, and ker $(\theta)$ represents the kernel of the function $\theta$. Let $m + n\sum_{j=1}^{7}(i_j) = c + d\left(\sum_{j=1}^{7}(i_j)\right)(x + y\left(\sum_{j=1}^{7}(i_j)\right)$, where $x$ and $y$ are two rational numbers. Since, $0 \equiv m + n\sum_{j=1}^{7}(i_j) = m - (cd^{-1})n$, $0 \equiv dm - cn$, which by

$$
\begin{aligned}
\left(x + y \sum_{j=1}^{7}(i_j)\right) &= \frac{\left(m + n\sum_{j=1}^{7}(i_j)\right)}{\left(c + d\sum_{j=1}^{7}(i_j)\right)} \\
&= \frac{cm + 7dn}{c^2 + 7d^2} + \left(\sum_{j=1}^{7}(i_j)\right)\frac{cn - dm}{c^2 + 7d^2}
\end{aligned}
$$

makes $y$ is an integer. Multiply $cd$ by equation $0 \equiv dm - cn$, which implies that $0 \equiv cd^2 m - c^2 dn$, which implies that $0 \equiv cm - c^2 d^{-2}dn$. from $(cd^{-1})^2 \equiv -7$, we have $0 \equiv cm + 7dn$, thus $x$ is also an integer. We conclude $< c + d\sum_{j=1}^{7}(i_j) > \supseteq Ker(\theta)$, which means that $< c + d\sum_{j=1}^{7}(i_j) > = Ker(\theta)$. Hence, it is proven that $O(K)/< c + d\sum_{j=1}^{7}(i_j) >$ is isomorphic to $\mathbb{Z}_{c^2+7d^2}$. $\square$

### 3.1. Residue Class of Octonion Integers

Let $O(K)_{\delta^k}$ be the residue class of $O(K)$ over modulo $\delta^k$, where $k$ is any positive integer and $\delta$ is an octonion prime integer. Conferring to the modulo function, $\sigma\colon Z_{p^k} \to O(K)_{\delta^k}$.

Defined in Equation (1):

$$o \to o - \left[ \frac{o\overline{\delta^k}}{\delta\overline{\delta^k}} \right] \delta^k \tag{1}$$

$O(K)_{\delta^k}$ is isomorphic to $Z_{p^k}$, where $p = \delta\overline{\delta}$ and $p$ is an odd prime integer. $\delta^k$ can be substituted by $\delta_1\delta_2\delta_3\ldots\delta_k$ in Equation (1), where $\delta_1, \delta_2, \delta_3, \ldots, \delta_k$ are different octonion integers. In Equation (1), $[.]$ represents the rounding of the real part and coefficient of the vector part of the octonion integer to the closest integer. A linear code of length $n$ over $O(K)_m$ of all $n$-tuples modulo spaces of $O(K)_m$ is $O(K)_m^n$. A cyclic code over octonion integers of length $n$ is a linear code of length $n$ by the following property:

$$(c_0, c_1, c_2, \ldots, c_{n-1}) \in C$$
$$\implies (c_{n-1}, c_0, \ldots, c_{n-2}) \in C$$

In the present case, we have a 1-1, and onto the map:

$$O(K)_m^n \to O(K)_m[x]/(x^n - 1),$$
$$(c_0, c_1, c_2, \ldots, c_{n-1}) \mapsto c_0 + c_1 x + c_2 x^2 + \ldots + c_{n-1}x^{n-1} + (x^n - 1) \tag{2}$$

To put it simply, we write: $c_0 + c_1 x + c_2 x^2 + \ldots + c_{n-1}x^{n-1}$ for $c_0 + c_1 x + c_2 x^2 + \ldots + c_{n-1}x^{n-1} + (x^n - 1)$. A non-empty set of $O(K)_m^n$ is a $O(K)_m$ cyclic code *iff* its image in Equation (2) is an ideal of $O(K)_m[x]/(x^n - 1)$.

**Definition 2.** *Let $\beta$, $\gamma \in O(K)_\delta$, $\alpha = \gamma - \beta = c + d\sum_{j=1}^{7}(i_j)$ be a prime octonion integer. Then, the Octonion Mannheim's (OM) Weight of $\alpha$ defined as*

$$W_{OM}(\alpha) = |c| + 7|d|$$

An Octonion Mannheim (OM) Distance $d_{OM}$ between $\beta$ and $\gamma$ is defined as

$$d_{OM}(\beta, \gamma) = W_{OM}(\alpha)$$

**Proposition 1.** *Let $\delta = c + d\sum_{j=1}^{7}(i_j)$ be a set of primes in $O(K)$, and let $p = c^2 + 7d^2$ be prime in $\mathbb{Z}$. If $O(K)_{\delta^2}^*$ is generated by $g$, then $g^{\frac{\varphi(p^2)}{2}} \equiv -1\left(mod\ \delta^2\right)$, where $\varphi$ represents the Euler phi function.*

**Proof.** If $N(\delta)$ is a prime integer in $\mathbb{Z}$, then the greatest common divisor of the real part and coefficient of the imaginary part of $\delta^2$ is 1. Then, $\mathbb{Z}_{P^2}$ is isomorphic to $O(K)_{\delta^2}$ (by Theorem 2). If $O(K)_{\delta^2}^*$ is generated by $g$, then $g, g^2, g^3, \ldots, g^{\varphi(p^2)}$ constitutes a reduced residue system modulo $\delta^2$ in $O(K)_{\delta^2}$. Thus, a positive integer $k$, such that $g^k \equiv -1\left(mod\ \delta^2\right)$, where $1 \le k \le \varphi(P^2)$. Hence, we concluded that $g^{2k} \equiv 1\left(mod\ \delta^2\right)$. Since $\varphi(p^2)|2k$ and $2 \le 2k \le 2\varphi(p^2)$, we take $\varphi(p^2) = k$ or $\varphi(p^2) = 2k$. If $\varphi(p^2)$ is $k$, then we should have $\delta^2|2$, but this is a contradiction of fact $N(\delta^2) > 2$. Hence, if $O(K)_{\delta^2}^*$ is generated by $g$ then $g^{\frac{\varphi(p^2)}{2}} \equiv -1\left(mod\ \delta^2\right)$, where $\varphi$ represents the Euler phi function. $\square$

The next Proposition is the generalization of Proposition 6.

**Proposition 2.** *Let $\delta_k = c_k + d_k \sum_{j=1}^{7}(i_j)$ be the different primes in $O(K)$ and $p_k = c_k^2 + 7d_k^2$ are distinct prime integers in $\mathbb{Z}$, where $k = 1, 2, 3, \ldots, m$. If $O(K)_{\delta^2}^*$ is generated by $g$ then $g^{\frac{\varphi(P^k)}{2}} \equiv -1\left(mod\ \delta^k\right)$.*

**Proof.** *If* $N(\delta)$ is a prime in $\mathbb{Z}$, then the greatest common divisor of the real part and coefficient of the imaginary part $\delta^k$ is 1. Then, $\mathbb{Z}_{p^k}$ is isomorphic to $O(K)_{\delta^k}$ (by Theorem 2). If $O(K)_{\delta^k}^*$ is generated by $g$, then $g, g^2, g^3, \ldots, g^{\varphi(p^k)}$ constitute a reduced residue system modulo $\delta^k$ in $O(K)_{\delta^k}$. Thus, a positive integer $s$, such that $g^s \equiv -1\left(mod\ \delta^k\right)$, where $1 \leq s \leq \varphi\left(P^k\right)$. Hence, we concluded thatinline-formula> $g^{2s} \equiv 1\left(mod\ \delta^k\right)$. Since $\varphi\left(P^k\right)\Big|2s$ and $2 \leq 2s \leq 2\varphi\left(P^k\right)$, we take $\varphi(P^2) = s$ or $\varphi\left(P^k\right) = 2s$. If $\varphi\left(P^k\right)$ is $s$, we should have $\delta^k\Big|2$, but this is a contradiction of fact $N(\delta^k) > 2$. $\square$

The next Theorem shows the cyclic code of length $n = \varphi(p^2)/2$ over the ring $O(K)_{\delta^2}$.

**Theorem 4.** *Let $\delta = c + d\sum_{j=1}^{7}(i_j)$ be a set of primes in $O(K)$ and $p = c^2 + 7d^2$ is a prime integer $\mathbb{Z}$, where $c,\ d \in \mathbb{Z}$. Then, there exists a $\frac{\varphi(p^2)}{2}$ length cyclic code over the ring $O(K)_{\delta^2}$.*

**Proof.** $O(K)_{\delta^2}^*$ has a generator since $\mathbb{Z}_{p^2} \cong O(K)_{\delta^2}$. Let $g$ be a generator. Then, $\varphi\left(p^2\right) = 1$. By Propositions 1 and 2; $g^{\varphi(p^2)/2} = -1$. Hence, $x^{\varphi(p^2)/2} + 1$ can be written

$$x^{\varphi(p^2)/2} + 1 = (x - g)Q(x)\left(mod\ \delta^2\right)(for\ x = g). \tag{3}$$

In this case, the Ideal of $O(K)_{\delta^2}[x]/< g^{\frac{\varphi(p^2)}{2}+1} >$ is $(x - g)$. Hence, cyclic code is generated. $\square$

If we chose $g(x)$ generator polynomial as a monic polynomial, then any row of generator matrix $G$ consists without zero divisors.

Propositions 3 and 4 showed the isomorphic relation between two or more residue classes of octonion integers.

**Proposition 3.** *Let $\delta_1 = c + d\sum_{j=1}^{7}(i_j)$ and $\delta_2 = a + b\sum_{j=1}^{7}(i_j)$ be the two prime integers in $O(K)$ and $p_1 = c^2 + 7d^2$, $p_2 = a^2 + 7b^2$ are two prime integers in $\mathbb{Z}$. Then, two elements of $O(K)_{\delta_1\delta_2}^*$, such that $e^{\varphi(p_2)} \equiv 1(mod\ \delta_1\delta_2)$ and $f^{\varphi(p_1)} \equiv 1(mod\ \delta_1\delta_2)$.*

**Proof.** Since the greatest common divisor of $P_1$ and $P_2$ is 1 in $\mathbb{Z}$, the greatest common divisor of $\delta_1$ and $\delta_2$ is 1 in $O(K)$. Using the simple algebraic concepts and the function (1),

$$\mathbb{Z}_{P_1} \cong O(K)_{\delta_1},\ \mathbb{Z}_{P_2} \cong O(K)_{\delta_2},\ and\ \mathbb{Z}_{P_1P_2} \cong O(K)_{\delta_1\delta_2}.$$

Furthermore, we get the following:

$$O(K)_{\delta_1\delta_2}^*(\delta_1) \cong \mathbb{Z}_{P_1P_2}^*(P_1) \cong \mathbb{Z}_{P_2}^* \cong O(K)_{\delta_2}^*,$$
$$O(K)_{\delta_1\delta_2}^*(\delta_2) \cong \mathbb{Z}_{P_1P_2}^*(P_2) \cong \mathbb{Z}_{P_1}^* \cong O(K)_{\delta_1}^*$$

Since $\delta_2$ is a prime octonion integer, by previous Proposition $O(K)_{\delta_2}^*$ is a cyclic group. Then, $O(K)_{\delta_2}^*$ has a generator.

Thus $O(K)_{\delta_1\delta_2}^*(\delta_1)$ has a generator too. Let $O(K)_{\delta_2}^*$ is generated by $\tau$. Then, $\tau^{\varphi(p_2)} \equiv 1(mod\ \delta_1\delta_2)$. In the same way, $O(K)_{\delta_1\delta_2}^*(\delta_2)$ has a generator. Suppose that $O(K)_{\delta_1\delta_2}^*(\delta_2)$ has a generator $f$. Then, $f^{\varphi(p_1)} \equiv 1(mod\ \delta_1\delta_2)$. $\square$

**Proposition 4.** *Let $\delta_k = c_k + d_k \sum_{j=1}^{7} (i_j)$ be the different primes in $O(K)$ and $p_k = c_k^2 + 7d_k^2$ are different prime integers in $\mathbb{Z}$, Then, $\tau_k$ be the element of $O(K)^*_{\delta_1 \delta_2 \dots \delta_k}$, such that $\tau_k^{\varphi(p_k)} \equiv 1 (mod\ \delta_1 \delta_2 \dots \delta_k)$, $k = 1, 2, 3, \dots, m$.*

**Proof.** It is proved by mathematical induction using Proposition 3. □

**Theorem 5.** *Let $\delta_1 = c + d \sum_{j=1}^{7} (i_j)$ and $\delta_2 = a + b \sum_{j=1}^{7} (i_j)$ be the two different prime integers in $O(K)$ and $p_1 = c^2 + 7d^2$, $p_2 = a^2 + 7b^2$ are two different odd prime integers in $\mathbb{Z}$. Then, there exist $\varphi(p_1)$ and $\varphi(p_2)$ lengths cyclic codes over $O(K)_{\delta_1 \delta_2}$.*

**Proof.** By Proposition 3, we can find an element of $O(K)_{\delta_1 \delta_2}$, such that $\tau^{\varphi(P_1)} \equiv 1(mod\ \delta_1 \delta_2)$. Thus, $x^{\varphi(P_1)} - 1$ factorizes the polynomial over $O(K)_{\delta_1 \delta_2}$ as $x^{\varphi(P_1)} - 1 = (x - \tau)D(x)(mod\ \delta_1 \delta_2)$. If we chose $g(x) = x - \tau$ generator polynomial, then $g(x)$ forms the generator matrix $G$, which consists of all elements of any rows without zero divisors. Thus, we get $\varphi(P_1)$ length code $C$, and it is generated by $x - \tau$. In the same techniques, we can get $\varphi(P_2)$ length code. □

**Remark 1.** *The cyclic code $C$ generated by $G$ consists of all linear combinations of the rows of the generator matrix $G$. So, the cardinality of cyclic codes over octonion integers is $p^k$. Additionally, the dimension of cyclic codes is $k$, which consists of all linearly independent rows of generator matrix $G$.*

*3.2. Decoding Procedure of Cyclic Codes for Error of One Octonion Mannheim Weight*

Let $r$ be the received vector, and $S$ be the syndrome with the parity check matrix $H$ and the transpose of the received vector $r$ as $S = Hr^t$. If the syndrome is zero, then there will be no error in the received vector during transmission, but if the syndrome is not zero, then an error will occur in the received vector during transmission. Let syndrome $S = Hr^t = \left[ \omega^l \right]$, $l \equiv q(mod\ n)$. It means an error occurred in a received vector at the $(q + 1)th$ place, and the error value is computed by $\frac{\omega^l}{\omega^q}$.

The following examples show the whole finding of all Theorems, Propositions, and Corollaries of this article.

Example for the residue class of octonion integers and encoding of cyclic codes over Octonion Integers: Let $\delta = 2 + \sum_{j=1}^{7} (i_j)$ and $n = \frac{\varphi(p^2)}{2}$. By using Theorem 3.5, $x^{55} + 1$ polynomial can be factorized over $O(K)_{\delta^2}$ as

$$x^{55} + 1 = (x - \omega)\left( x^{54} + \omega x^{53} + \omega^2 x^{52} + \dots + \omega^{53} x + \omega^{54} \right), \text{Where } \omega = -1 - \sum_{j=1}^{7} (i_j).$$

If we take $g(x) = x - \omega$ generator polynomial, then generator matrix $(G)$ is

$$G = [a_{ij}]_{54 \times 55}$$

$$G = \begin{pmatrix} -1 - \sum\limits_{j=1}^{7} (i_j) & 1 & 0 & 0 \cdots 0 & 0 \\ 0 & -1 - \sum\limits_{j=1}^{7} (i_j) & 1 & 0 \cdots 0 & 0 \\ 0 & 0 & -1 - \sum\limits_{j=1}^{7} (i_j) & 1 \cdots 0 & 0 \\ \vdots & \vdots & \vdots & \vdots & \vdots \\ & & 0\,0\,0\,0 \cdots 1\,0 & & \\ & & 0\,0\,0\,0 \cdots -1 - \sum\limits_{j=1}^{7} (i_j)\ 1 & & \end{pmatrix}$$

The cyclic code generated by $G$ is consists of all linear combinations of the rows of generator matrix $G$. Thus, the cardinality of cyclic codes over octonion integers is $p^k = 11^{54}$. Additionally, the dimension of cyclic codes is $k = 54$, which consists of all linearly independent rows of generator matrix $G$.

Example for Error corrections of octonion Mannheim weight one: Let $\delta = 2 + \sum_{j=1}^{7}(i_j)$, and $n = \frac{\varphi(p^2)}{2}$. By using Theorem 3.5, $x^{55} + 1$ polynomial can be factorized over $O(K)_{\delta^2}$ as:

$x^{55} + 1 = (x - \omega)(x^{54} + \omega x^{53} + \omega^2 x^{52} + \ldots + \omega^{53} x + \omega^{54}) = g(x) * h(x)$, $h(x) = x^{54} + \omega x^{53} + \omega^2 x^{52} + \ldots + \omega^{53} x + \omega^{54}$ is the check polynomial. Parity check matrix $H$ is defined by check polynomial as

$$H = \left( \omega^{56}\ \omega^{57}\ \omega^{58}\ \omega^{59}\ \cdots\ \omega^{109}\ 1 \right)_{1 \times 55}$$
$$= \left( -1 + \sum_{j=1}^{7}(i_j)\ \ 12 - 6\sum_{j=1}^{7}(i_j)\ \ 12 + 47\sum_{j=1}^{7}(i_j)\ \ \cdots\ \ 4 - 51\sum_{j=1}^{7}(i_j)\ \ 1 \right)_{1 \times 55}$$

Let $r = (1\ 1 + \sum_{j=1}^{7}(i_j)\ 0\ \ldots\ -1\ 0)_{1 \times 55}$ be a received vector during transmission.

Then, the syndrome $S = Hr^t = \omega^{108}$ (See Table 1), and the error occurred in the received vector $r$ at position $108 \equiv 53 (mod\ 55)$, so that the error value $= e = \frac{\omega^{108}}{\omega^{53}} = -1$ (from Table 1). Thus, the octonion mannhein weight of error is 1. Hence, the corrected code word is

$$c = r - e = (1\ 1 + \sum_{j=1}^{7}(i_j)\ 0\ \ldots\ 0\ 0)_{1 \times 55}. \tag{4}$$

**Table 1.** Cyclic group over the residue class of octonion integers generated by $\omega$.

| $u$ | $\omega^u$ |
|---|---|
| 1 | $1 - i_1 - i_2 - i_3 - i_4 - i_5 - i_6 - i_7$ |
| 2 | $-12 + 6i_1 + 6i_2 + 6i_3 + 6i_4 + 6i_5 + 6i_6 + 6i_7$ |
| 3 | $-12 - 47i_1 - 47i_2 - 47i_3 - 47i_4 - 47i_5 - 47i_6 - 47i_7$ |
| 4 | $-14 + 13i_1 + 13i_2 + 13i_3 + 13i_4 + 13i_5 + 13i_6 + 13i_7$ |
| 5 | $-7 + 18i_1 + 18i_2 + 18i_3 + 18i_4 + 18i_5 + 18i_6 + 18i_7$ |
| 6 | $-56 - 24i_1 - 24i_2 - 24i_3 - 24i_4 - 24i_5 - 24i_6 - 24i_7$ |
| 7 | $-40 - 52i_1 - 52i_2 - 52i_3 - 52i_4 - 52i_5 - 52i_6 - 52i_7$ |
| 8 | $-30 + 54i_1 + 54i_2 + 54i_3 + 54i_4 + 54i_5 + 54i_6 + 54i_7$ |
| 9 | $9 + 52i_1 + 52i_2 + 52i_3 + 52i_4 + 52i_5 + 52i_6 + 52i_7$ |
| 10 | $51 - 52i_1 - 52i_2 - 52i_3 - 52i_4 - 52i_5 - 52i_6 - 52i_7$ |
| 11 | $-36 + 52i_1 + 52i_2 + 52i_3 + 52i_4 + 52i_5 + 52i_6 + 52i_7$ |
| 12 | $42 - 55i_1 - 55i_2 - 55i_3 - 55i_4 - 55i_5 - 55i_6 - 55i_7$ |
| 13 | $40 - 43i_1 - 43i_2 - 43i_3 - 43i_4 - 43i_5 - 43i_6 - 43i_7$ |
| 14 | $54 - 19i_1 - 19i_2 - 19i_3 - 19i_4 - 19i_5 - 19i_6 - 19i_7$ |
| 15 | $54 + 32i_1 + 32i_2 + 32i_3 + 32i_4 + 32i_5 + 32i_6 + 32i_7$ |
| 16 | $-60 - 15i_1 - 15i_2 - 15i_3 - 15i_4 - 15i_5 - 15i_6 - 15i_7$ |
| 17 | $-8 - 3i_1 - 3i_2 - 3i_3 - 3i_4 - 3i_5 - 3i_6 - 3i_7$ |
| 18 | $44 + 25i_1 + 25i_2 + 25i_3 + 25i_4 + 25i_5 + 25i_6 + 25i_7$ |
| 19 | $42i_1 + 42i_2 + 42i_3 + 42i_4 + 42i_5 + 42i_6 + 42i_7$ |
| 20 | $2 + 28i_1 + 28i_2 + 28i_3 + 28i_4 + 28i_5 + 28i_6 + 28i_7$ |
| 21 | $-7 + 17i_1 + 17i_2 + 17i_3 + 17i_4 + 17i_5 + 17i_6 + 17i_7$ |
| 22 | $52 - 17i_1 - 17i_2 - 17i_3 - 17i_4 - 17i_5 - 17i_6 - 17i_7$ |

**Table 1.** *Cont.*

| $u$ | $\omega^u$ |
|:---:|:---:|
| 23 | $-40 + 16i_1 + 16i_2 + 16i_3 + 16i_4 + 16i_5 + 16i_6 + 16i_7$ |
| 24 | $-32 - 7i_1 - 7i_2 - 7i_3 - 7i_4 - 7i_5 - 7s_6 - 7s_7$ |
| 25 | $16 + 57i_1 + 57i_2 + 57i_3 + 57i_4 + 57i_5 + 57i_6 + 57i_7$ |
| 26 | $2 + 27i_1 + 27i_2 + 27i_3 + 27i_4 + 27i_5 + 27i_6 + 27i_7$ |
| 27 | $-20 + 24i_1 + 24i_2 + 24i_3 + 24i_4 + 24i_5 + 24i_6 + 24i_7$ |
| 28 | $52 + 51i_1 + 51i_2 + 51i_3 + 51i_4 + 51i_5 + 51i_6 + 51i_7$ |
| 29 | $-42 - 45i_1 - 45i_2 - 45i_3 - 45i_4 - 45i_5 - 45i_6 - 45i_7$ |
| 30 | $-25 - 2i_1 - 2i_2 - 2i_3 - 2i_4 - 2i_5 - 2i_6 - 2i_7$ |
| 31 | $-33 + 15i_1 + 15i_2 + 15i_3 + 15i_4 + 15i_5 + 15i_6 + 15i_7$ |
| 32 | $18 - i_1 - i_2 - i_3 - i_4 - i_5 - i_6 - i_7$ |
| 33 | $-10 + 9i_1 + 9i_2 + 9i_3 + 9i_4 + 9i_5 + 9i_6 + 9i_7$ |
| 34 | $-30 + 49i_1 + 49i_2 + 49i_3 + 49i_4 + 49i_5 + 49i_6 + 49i_7$ |
| 35 | $-28i_1 - 28i_2 - 28i_3 - 28i_4 - 28i_5 - 28i_6 - 28i_7$ |
| 36 | $9 - 19i_1 - 19i_2 - 19i_3 - 19i_4 - 19i_5 - 19i_6 - 19i_7$ |
| 37 | $45 + 29i_1 + 29i_2 + 29i_3 + 29i_4 + 29i_5 + 29i_6 + 29i_7$ |
| 38 | $16 + 11i_1 + 11i_2 + 11i_3 + 11i_4 + 11i_5 + 11i_6 + 11i_7$ |
| 39 | $4 + 33i_1 + 33i_2 + 33i_3 + 33i_4 + 33i_5 + 33i_6 + 33i_7$ |
| 40 | $-30 - 21i_1 - 21i_2 - 21i_3 - 21i_4 - 21i_5 - 21i_6 - 21i_7$ |
| 41 | $7 + 46i_1 + 46i_2 + 46i_3 + 46i_4 + 46i_5 + 46i_6 + 46i_7$ |
| 42 | $-60 - 7i_1 - 7i_2 - 7i_3 - 7i_4 - 7i_5 - 7i_6 - 7i_7$ |
| 43 | $9 + 57i_1 + 57i_2 + 57i_3 + 57i_4 + 57i_5 + 57i_6 + 57i_7$ |
| 44 | $60 + 28i_1 + 28i_2 + 28i_3 + 28i_4 + 28i_5 + 28i_6 + 28i_7$ |
| 45 | $6 + 19i_1 + 19i_2 + 19i_3 + 19i_4 + 19i_5 + 19i_6 + 19i_7$ |
| 46 | $-39 - 32i_1 - 32i_2 - 32i_3 - 32i_4 - 32i_5 - 32i_6 - 32i_7$ |
| 47 | $-55 + 12i_1 + 12i_2 + 12i_3 + 12i_4 + 12i_5 + 12i_6 + 12i_7$ |
| 48 | $-31 + 26i_1 + 26i_2 + 26i_3 + 26i_4 + 26i_5 + 26i_6 + 26i_7$ |
| 49 | $17 + 34i_1 + 34i_2 + 34i_3 + 34i_4 + 34i_5 + 34i_6 + 34i_7$ |
| 50 | $-13 - 29i_1 - 29i_2 - 29i_3 - 29i_4 - 29i_5 - 29i_6 - 29i_7$ |
| 51 | $-8 - 11i_1 - 11i_2 - 11i_3 - 11i_4 - 11i_5 - 11i_6 - 11i_7$ |
| 52 | $60 - 35i_1 - 35i_2 - 35i_3 - 35i_4 - 35i_5 - 35i_6 - 35i_7$ |
| 53 | $17 + 39i_1 + 39i_2 + 39i_3 + 39i_4 + 39i_5 + 39i_6 + 39i_7$ |
| 54 | $-4 + 51i_1 + 51i_2 + 51i_3 + 51i_4 + 51i_5 + 51i_6 + 51i_7$ |
| 55 | $-1$ |
| 56 | $-1 + i_1 + i_2 + i_3 + i_4 + i_5 + i_6 + i_7$ |
| 57 | $+12 - 6i_1 - 6i_2 - 6i_3 - 6i_4 - 6i_5 - 6i_6 - 6i_7$ |
| 58 | $12 + 47i_1 + 47i_2 + 47i_3 + 47i_4 + 47i_5 + 47i_6 + 47i_7$ |
| 59 | $14 - 13i_1 - 13i_2 - 13i_3 - 13i_4 - 13i_5 - 13i_6 - 13i_7$ |
| 60 | $7 - 18i_1 - 18i_2 - 18i_3 - 18i_4 - 18i_5 - 18i_6 - 18i_7$ |
| 61 | $56 + 24i_1 + 24i_2 + 24i_3 + 24i_4 + 24i_5 + 24i_6 + 24i_7$ |
| 62 | $40 + 52i_1 + 52i_2 + 52i_3 + 52i_4 + 52i_5 + 52i_6 + 52i_7$ |
| 63 | $30 - 54i_1 - 54i_2 - 54i_3 - 54i_4 - 54i_5 - 54i_6 - 54i_7$ |

**Table 1.** *Cont.*

| $u$ | $\omega^u$ |
|---|---|
| 64 | $-9 - 52i_1 - 52i_2 - 52i_3 - 52i_4 - 52i_5 - 52i_6 - 52i_7$ |
| 65 | $-51 + 52i_1 + 52i_2 + 52i_3 + 52i_4 + 52i_5 + 52i_6 + 52i_7$ |
| 66 | $36 - 52i_1 - 52i_2 - 52i_3 - 52i_4 - 52i_5 - 52i_6 - 52i_7$ |
| 67 | $-42 + 55i_1 + 55i_2 + 55i_3 + 55i_4 + 55i_5 + 55i_6 + 55i_7$ |
| 68 | $-40 + 43i_1 + 43i_2 + 43i_3 + 43i_4 + 43i_5 + 43i_6 + 43i_7$ |
| 69 | $-54 + 19i_1 + 19i_2 + 19i_3 + 19i_4 + 19i_5 + 19i_6 + 19i_7$ |
| 70 | $-54 - 32i_1 - 32i_2 - 32i_3 - 32i_4 - 32i_5 - 32i_6 - 32i_7$ |
| 71 | $60 + 15i_1 + 15i_2 + 15i_3 + 15i_4 + 15i_5 + 15i_6 + 15i_7$ |
| 72 | $8 + 3i_1 + 3i_2 + 3i_3 + 3i_4 + 3i_5 + 3i_6 + 3i_7$ |
| 73 | $-44 - 25i_1 - 25i_2 - 25i_3 - 25i_4 - 25i_5 - 25i_6 - 25i_7$ |
| 74 | $-42i_1 - 42i_2 - 42i_3 - 42i_4 - 42i_5 - 42i_6 - 42i_7$ |
| 75 | $-2 - 28i_1 - 28i_2 - 28i_3 - 28i_4 - 28i_5 - 28i_6 - 28i_7$ |
| 76 | $7 - 17i_1 - 17i_2 - 17i_3 - 17i_4 - 17i_5 - 17i_6 - 17i_7$ |
| 77 | $-52 + 17i_1 + 17i_2 + 17i_3 + 17i_4 + 17i_5 + 17i_6 + 17i_7$ |
| 78 | $40 - 16i_1 - 16s_2 - 16i_3 - 16i_4 + 16i_5 - 16i_6 - 16i_7$ |
| 79 | $32 + 7i_1 + 7i_2 + 7i_3 + 7i_4 + 7i_5 + 7i_6 + 7i_7$ |
| 80 | $-16 - 57i_1 - 57i_2 - 57i_3 - 57i_4 - 57i_5 - 57i_6 - 57i_7$ |
| 81 | $-2 - 27i_1 - 27i_2 - 27i_3 - 27i_4 - 27i_5 - 27i_6 - 27i_7$ |
| 82 | $20 - 24i_1 - 24i_2 - 24i_3 - 24i_4 - 24i_5 - 24i_6 - 24i_7$ |
| 83 | $-52 - 51i_1 - 51i_2 - 51i_3 - 51i_4 - 51i_5 - 51i_6 - 51i_7$ |
| 84 | $42 + 45i_1 + 45i_2 + 45i_3 + 45i_4 + 45i_5 + 45i_6 + 45i_7$ |
| 85 | $25 + 2i_1 + 2i_2 + 2i_3 + 2i_4 + 2i_5 + 2i_6 + 2i_7$ |
| 86 | $33 - 15i_1 - 15i_2 - 15i_3 - 15i_4 - 15i_5 - 15i_6 - 15i_7$ |
| 87 | $-18 + i_1 + i_2 + i_3 + i_4 + i_5 + i_6 + i_7$ |
| 88 | $10 - 9i_1 - 9i_2 - 9i_3 - 9i_4 - 9i_5 - 9i_6 - 9i_7$ |
| 89 | $30 - 49i_1 - 49i_2 - 49i_3 - 49i_4 - 49i_5 - 49i_6 - 49i_7$ |
| 90 | $28i_1 + 28i_2 + 28i_3 + 28i_4 + 28i_5 + 28i_6 + 28i_7$ |
| 91 | $-9 + 19i_1 + 19i_2 + 19i_3 + 19i_4 + 19i_5 + 19i_6 + 19i_7$ |
| 92 | $-45 - 29i_1 - 29i_2 - 29i_3 - 29i_4 - 29i_5 - 29i_6 - 29i_7$ |
| 93 | $-16 - 11i_1 - 11i_2 - 11i_3 - 11i_4 - 11i_5 - 11i_6 - 11i_7$ |
| 94 | $-4 - 33i_1 - 33i_2 - 33i_3 - 33i_4 - 33i_5 - 33i_6 - 33i_7$ |
| 95 | $30 + 21i_1 + 21i_2 + 21i_3 + 21i_4 + 21i_5 + 21i_6 + 21i_7$ |
| 96 | $-4 - 46i_1 - 46i_2 - 46i_3 - 46i_4 - 46i_5 - 46i_6 - 46i_7$ |
| 97 | $60 + 7i_1 + 7i_2 + 7i_3 + 7i_4 + 7i_5 + 7i_6 + 7i_7$ |
| 98 | $-9 - 57i_1 - 57i_2 - 57i_3 - 57i_4 - 57i_5 - 57i_6 - 57i_7$ |
| 99 | $-60 - 28i_1 - 28i_2 - 28i_3 - 28i_4 - 28i_5 - 28i_6 - 28i_7$ |
| 100 | $-6 - 19i_1 - 19i_2 - 19i_3 - 19i_4 - 19i_5 - 19i_6 - 19i_7$ |
| 101 | $39 + 32i_1 + 32i_2 + 32i_3 + 32i_4 + 32i_5 + 32i_6 + 32i_7$ |
| 102 | $55 - 12i_1 - 12i_2 - 12i_3 - 12i_4 - 12i_5 - 12i_6 - 12i_7$ |
| 103 | $31 - 26i_1 - 26i_2 - 26i_3 - 26i_4 - 26i_5 - 26i_6 - 26i_7$ |
| 104 | $-17 - 34i_1 - 34i_2 - 34i_3 - 34i_4 - 34i_5 - 34i_6 - 34i_7$ |

**Table 1.** *Cont.*

| $u$ | $\omega^u$ |
|---|---|
| 105 | $13 + 29i_1 + 29i_2 + 29i_3 + 29i_4 + 29i_5 + 29i_6 + 29i_7$ |
| 106 | $8 + 11i_1 + 11i_2 + 11i_3 + 11i_4 + 11i_5 + 11i_6 + 11i_7$ |
| 107 | $-60 + 35s_1 + 35s_2 + 35s_3 + 35s_4 + 35s_5 + 35s_6 + 35s_7$ |
| 108 | $-17 - 39i_1 - 39i_2 - 39i_3 - 39i_4 - 39i_5 - 39i_6 - 39i_7$ |
| 109 | $4 - 51i_1 - 51i_2 - 51i_3 - 51i_4 - 51i_5 - 51i_6 - 51i_7$ |
| 110 | $1$ |

## 4. Comparison

In [15], the author presented cyclic codes over the residue class of quaternion integer $H(K)_\pi$, $\pi = a + b\sum_{j=1}^{3}(i_j)$ with length $n = \frac{\varphi(p^2)}{2} = \frac{p(p-1)}{2}$, where $p = \pi\overline{\pi} = a^2 + 3b^2$ and dimension $k$. In the present article, the author presented cyclic codes over the residue class of octonion integer $O(K)_\pi$, $\pi = a + b\sum_{j=1}^{7}(i_j)$ with different length $n_1 = \frac{\varphi(p^2)}{2} = \frac{p(p-1)}{2}$," where $p = \pi\overline{\pi} = a^2 + 7b^2$ with dimension $k$. The cardinality of code $C$ is $|C| = p^k$, the code rate $(R)$ for quaternion integers $R = \frac{k}{n}$ and octonion integers $R = \frac{k}{n_1}$ are given in the following Tables 2 and 3, Figures 1 and 2.

**Table 2.** Primes, length, cardinality, and code rate (R) of code over quaternion integers.

| $a$ | $b$ | $\pi = a + b\sum_{j=1}^{3}(i_j)$ | $p = \pi\overline{\pi}$ | $n = \frac{\varphi(p^2)}{2}$ | $k$ | $\lvert C\rvert = p^k$ | $R = \frac{k}{n}$ |
|---|---|---|---|---|---|---|---|
| 2 | 1 | $2 + 1\sum_{j=1}^{3}(i_j)$ | 7 | 21 | 20 | $6^{20}$ | $\frac{20}{21}$ |
| 1 | 2 | $1 + 2\sum_{j=1}^{3}(i_j)$ | 13 | 78 | 77 | $13^{77}$ | $\frac{77}{78}$ |
| 2 | 3 | $2 + 3\sum_{j=1}^{3}(i_j)$ | 31 | 465 | 464 | $31^{464}$ | $\frac{464}{465}$ |
| 2 | 5 | $2 + 5\sum_{j=1}^{3}(i_j)$ | 79 | 3081 | 3080 | $79^{3080}$ | $\frac{3080}{3081}$ |

**Table 3.** Primes, length, cardinality, and code rate (R) of code over octonion integers.

| $a$ | $b$ | $\pi = a + b\sum_{j=1}^{7}(i_j)$ | $p = \pi\overline{\pi}$ | $n_1 = \frac{\varphi(p^2)}{2}$ | $k$ | $\lvert C\rvert = p^k$ | $R = \frac{k}{n}$ |
|---|---|---|---|---|---|---|---|
| 2 | 1 | $2 + 1\sum_{j=1}^{7}(i_j)$ | 11 | 55 | 54 | $11^{54}$ | $\frac{54}{55}$ |
| 1 | 2 | $1 + 2\sum_{j=1}^{7}(i_j)$ | 29 | 406 | 405 | $29^{405}$ | $\frac{405}{406}$ |
| 2 | 3 | $2 + 3\sum_{j=1}^{7}(i_j)$ | 67 | 2211 | 2210 | $67^{2211}$ | $\frac{2210}{2211}$ |
| 2 | 5 | $2 + 5\sum_{j=1}^{7}(i_j)$ | 179 | 15,931 | 15,930 | $179^{15,930}$ | $\frac{15,930}{15,931}$ |

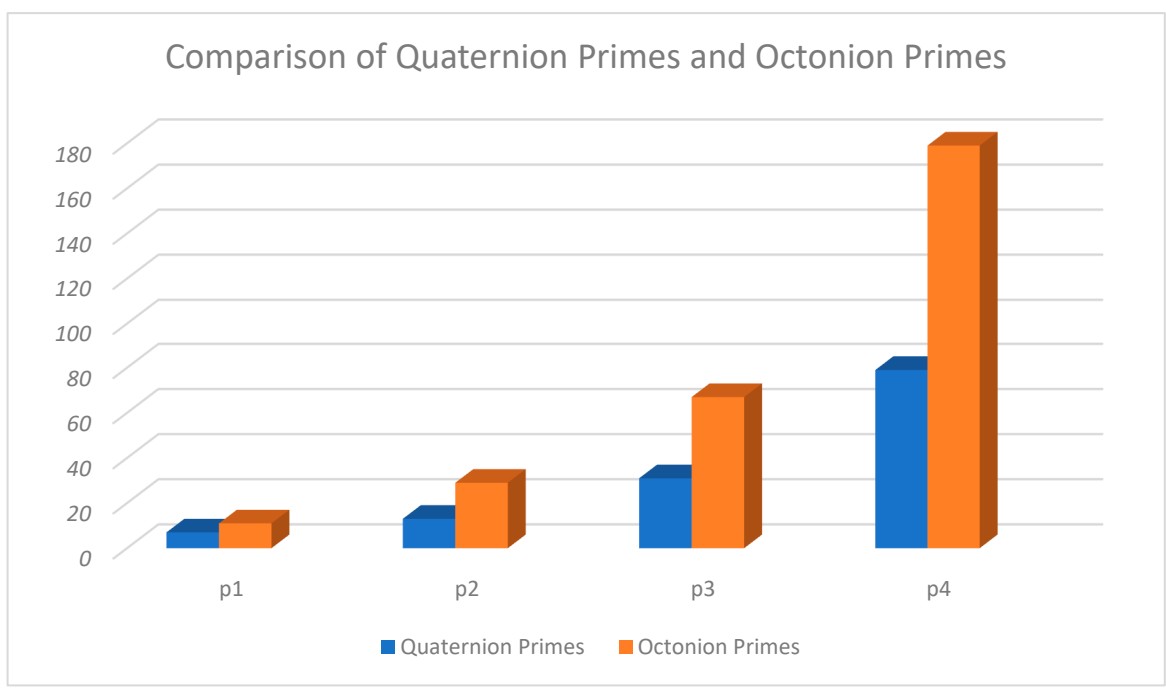

**Figure 1.** Quaternion Primes vs. Octonion Primes.

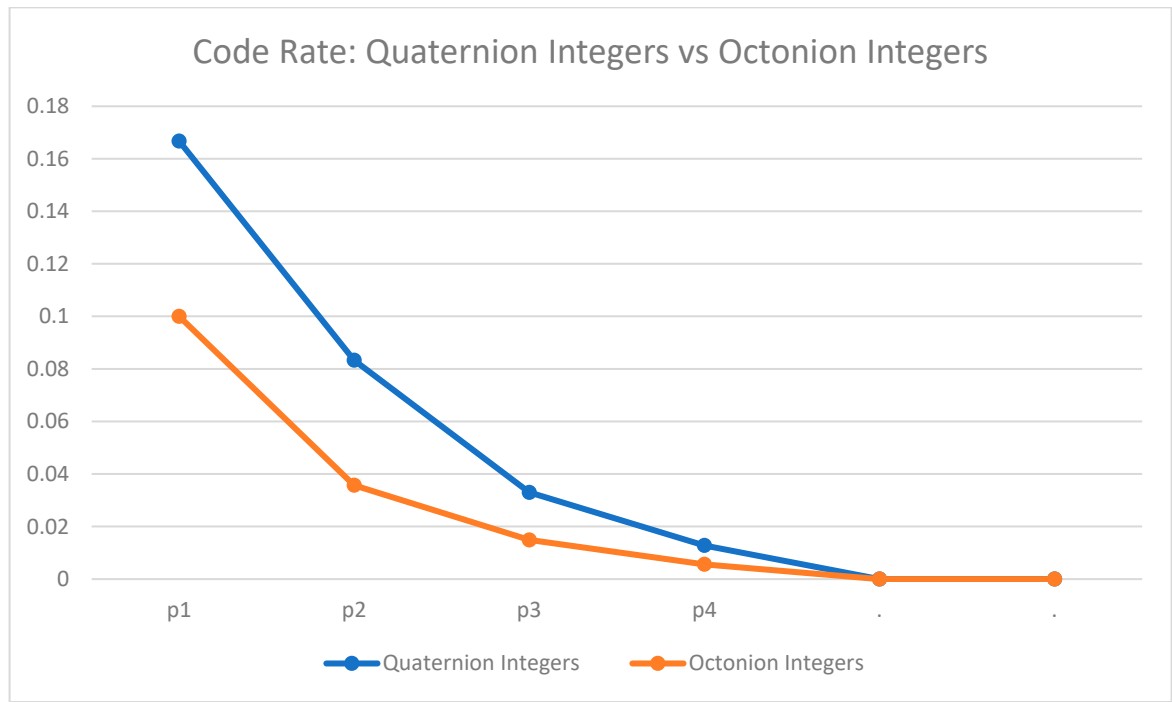

**Figure 2.** Quaternion integers Code rate vs. Octonion integers Code rate.

Tables 2 and 3, Figures 1 and 2 for the same *a* and *b* of both quaternion and octonion integers; prime, length, and cardinality of code slightly increased but code rate of code slightly decreased. If length of the cyclic codes slightly increased with the dimension *k*, then the transmission will be slightly slow and the error correction capability of cyclic codes over the residue class of octonion integers will be better as compared to the cyclic codes over the residue class of quaternion integers.

## 5. Conclusions

In this paper, we had generalized the results of [15], octonion integers, fundamental theorems related to octonion integers, cyclic codes, and error correction of cyclic codes over the residue class of octonion integers with respect to the octonion Mannheim distance. If the prime, length and cardinality of cyclic codes slightly increased with dimension $k$, then the transmission will be slightly slow, and the error correction capability of cyclic codes over the residue class of octonion integers will be better as compared to the cyclic codes over the residue class of quaternion integers.

Furthermore, the error correction and detection of cyclic codes over octonion integers may be extended to the error correction and detection of cyclic codes over sedenion integers.

**Author Contributions:** Conceptualization, M.S.; Supervisor, T.S.; Investigation, M.S., T.S., R.-J.S., Z.E.S.A. and O.S.D.; writing—review and editing, M.S., T.S., R.-J.S., Z.E.S.A. and O.S.D. All authors have read and agreed to the published version of the manuscript.

**Funding:** This work was supported by Universidad Pedagógica y Tecnológica de Colombia SGI 3334.

**Institutional Review Board Statement:** Not applicable.

**Informed Consent Statement:** Not applicable.

**Data Availability Statement:** Not applicable.

**Acknowledgments:** The authors thank the collaboration of all volunteers who participated in data collection.

**Conflicts of Interest:** The authors declare no conflict of interest.

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
