# Peer review of "Fundamental Results of Cyclic Codes over Octonion Integers and Their Decoding Algorithm"

_computation, doi:10.3390/computation10120219_

Round 1

Reviewer 1 Report

It is not real work, and the suitable title picture is not their in-depth analyzer for the Codes are studied by various scientific disciplines such as information theory, electrical engineering, mathematics, linguistics, and computer science to design efficient and reliable data transmission methodsComparison of primes, lengths, cardinality, dimension, and code rate will be discussed So, my succession has rejected this script. Thank you for giving those opportunists that journal and my review of this manuscript, which not only assisted me in reaching my decision but also enabled the author(s) to disseminate their work at the highest possible quality.

Title: Cyclic Codes over Octonion Integers: Encoding and Decoding 3 Algorithms

Journal: Computation

Manuscript ID:  Manuscript ID - computation-1981324

Authors are discussed in Coding theory is the study of the properties of codes and their respective fitness for specific applications. Codes are used for data compression, cryptography, error detection, error correction, data transmission, and data storage. But here we are discussing the highlight point not mentioned, usually discussing the same model several researchers already published in these ideas. It is not real work, and the suitable title picture is not their in-depth analyzer for the Codes are studied by various scientific disciplines such as information theory, electrical engineering, mathematics, linguistics, and computer science to design efficient and reliable data transmission methodsComparison of primes, lengths, cardinality, dimension, and code rate will be discussed So, my succession has rejected this script. Thank you for giving those opportunists that journal and my review of this manuscript, which not only assisted me in reaching my decision but also enabled the author(s) to disseminate their work at the highest possible quality.

However, I cannot recommend the manuscript for publication because of major flaws in the text presentation and discussions that compromise the understanding of the overall text. The confusing figures, bad presentation of the data and the poor level of English make me unable to follow the experimental procedure and the discussions in multiple sections.   If the methods and conclusions cannot be followed properly, it is not possible to evaluate and replicate the reported findings.

Therefore, I recommend that the text should be rejected.

Author Response

1) I have changed the Article Title as:

Fundamental Results of Cyclic Codes over Octonion Integers and Their Decoding Algorithm  

2) The author contributed the fundamental results of octonion integers, cyclic codes, and their decoding algorithm for single error correction of octonion Manheim weight one. 

3) This work is done by using different lengths and different parameters. 

For example, some authors have done this work by Quaternion integers with length n=(p-1)/2, and some have done cyclic codes over octonion integers by using length n=(p-1)/6, n=(p-1)/2, etc. 

But in this article, I have done fundamental results of cyclic codes and their decoding algorithm for length n=p-1. 

4) All other corrections are corrected and highlighted in the revision file of the manuscript.

Reviewer 2 Report

Codes are studied by scientific disciplines such as information theory, electrical engineering, mathematics, linguistics, and computer science. The main idea is the design of efficient and reliable data transmission methods. Many works (for instance [4, 5, 8, 11, 13, 14, 15, 22] in the manuscript) have discussed codes over finite fields, Gaussian integers, Hurwitz integers, quaternion integers, etc. In this article, the authors generalize the results of [15], cyclic codes over quaternion integer rings are extended to the case of Octonion integer rings concerning the Octonion Mannheim distance. They established fundamental theorems related to octonion integers, encoding, and decoding of cyclic codes over the residue class of octonion integers with respect to the octonion Mannheim distance. Particularly, Theorem 4 shows how to construct cyclic codes over these finite rings, at the end decoding of cyclic codes over octonion integers. Finally, a comparison between the quaternion and octonion integers with respect to primes, lengths, cardinality, dimension, and code rate has been discussed (in Table 2 and Table 3 of the manuscript). According to the manuscript, if the prime, length, and cardinality of cyclic codes slightly increased with dimension ? then the transmission will be slightly slow and the error correction capability of cyclic codes over the residue class of octonion integers will be better as compared to the cyclic codes over the residue class of quaternion integers. In conclusion, I consider that this manuscript deserves to be published in COMPUTATION MDPI for its importance in the applied field of algebra.

Author Response

Thanks for your cooperative response. 

Reviewer 3 Report

The authors develop encoding and decoding algorithms of cyclic codes over the residue class of octonion integers with respect to the octonion Mannheim distance. The results show that the prime, length and cardinality slightly increase with dimension and the error correction capability is better as compared to the cyclic codes over the residue class of quaternion integers.

Author Response

I have corrected it. Thanks for your cooperative response.

Round 2

Reviewer 1 Report

Publish